# Reflections on Rotational Osteotomies around the Patellofemoral Joint

**DOI:** 10.3390/jcm10030474

**Published:** 2021-01-27

**Authors:** Roland M. Biedert

**Affiliations:** Orthopaedic Surgery & Sports Traumatology, SportsClinic#1, Wankdorf Center, Papiermühlestrasse 73, CH-3014 Bern, Switzerland; biedert@sportsclinicnumber1.ch; Tel.: +41-31-356-11-11; Fax: +41-31-356-11-12

**Keywords:** femoral torsion, increased femoral antetorsion, rotational osteotomy, tibial torsion, patellofemoral joint, hip

## Abstract

Torsional abnormalities of the femur represent a significant risk factor for patellar instability or patellofemoral complaints. Although their clinical implication has been demonstrated, there is still a debate going on about different aspects. These include, especially, the various methods of measurements with a wide range of physiologic values, the indication or clear recommendation for surgical correction, and the site of the rotational osteotomy. Nevertheless, good subjective and objective functional results were reported after femoral rotational osteotomies. This is mostly not a review of the literature, but a collection of personal thoughts and observations.

## 1. Introduction

Why should we consider performing a rotational osteotomy at all?

This question is controversially debated in expert circles and no equal opinion exists. There is still a general lack of knowledge on this theme. Nevertheless, increased interest and the higher number of studies published in the latest literature underline the importance of this topic. Various factors became strong arguments in the last 30 years to consider rotational osteotomies. Accordingly, a profound discussion is raised and there is a strong need to assess different perspectives in detail.

Femoral torsion (FT) is a significant parameter in knee and hip disorders. Therefore, both joints must be integrated into the complete assessment.

The patellofemoral tracking is influenced by the complex interaction of the skeletal geometry, soft tissues, and neuromuscular control [1]. The patellofemoral and femorotibial joints act synergistically and ensure undisturbed movements in the knee joint. Under normal conditions, the high loading and shearing forces acting on the knee and the associated soft tissues are well controlled and optimally distributed. Pathologic limb factors may change these interactions and cause maltracking of the extensor mechanism and along the entire leg between the hip and foot. Torsional deformities influence the patellofemoral kinematics and cause altered vectors and forces acting on the patellofemoral joint (PFJ). This may lead to patellar instability, cartilage failure with secondary osteoarthritis due to increased patellofemoral stress, and musculotendinous insufficiency [1,2]. Therefore, they should be depicted and addressed for the appropriate treatment. However, rotational deformities of the lower limb are often missed or ignored due to difficulties in their assessment and the necessary treatment.

In recent years, we have observed an increasing number of patients with combined hip and patellofemoral problems. In some cases, the MRIs of hip and knee showed normal conditions. The only pathologic finding was an overall increase in femoral antetorsion. The clinical examination of the hip confirmed the pathologic finding of an increase in internal rotation. Complaints on both locations disappeared after supracondylar external rotation osteotomy.

In addition, torsional abnormalities were recently also integrated into the concept of femoroacetabular impingement, as they affect the impingement-free hip range of motion [3]. Excessively high FT may cause posterior extraarticular ischiofemoral impingement (conflict between ischium and lesser or major trochanter) with posterior and gluteal pain. Decreased FT may cause an anterior impingement resulting in anterior groin pain with labrum lesions (subluxation movements) and snapping psoas tendon (hyperextension of the tendon). By paying more attention to this, we have observed many patients suffering from patellofemoral complaints and the corresponding hip pain. Accordingly, both joints must be assessed in detail for a precise diagnosis.

Considering these aspects of torsional abnormalities, numerous questions arise: Which criteria and parameters are helpful for a better understanding of this subject? What are the points of restraint or hesitation? Which observations from daily practice are valuable? What subjects are/remain unclear?

## 2. Clinical Evaluation

Which observations should put us on track?

Torsional deformities of the femur were recognized as a possible cause of patellofemoral problems. A long history with ongoing symptoms, unsuccessful conservative treatment over a long period, functional disability, and the clinical examination are important factors for a primary diagnosis.

### 2.1. General Examination

The clinical diagnosis of abnormal FT may be difficult and a structured evaluation is needed. At first, the specific assessment of the knee with focus on the PFJ and the soft tissue structures is performed (instability tests of the patella, tightness, patella position, patella height, muscle conditions, contractures). Secondly, a clinical evaluation consists of the measurement of the hip range of movement in a prone and supine position. In patients presenting with pathologic high femoral antetorsion, increased internal and decreased external hip rotation in 90° of hip flexion is documented, often combined with a positive posterior extraarticular impingement sign with gluteal pain. Therefore, both anterior and posterior impingement tests (FABER test) should be performed [3].

In addition to the clinical examination of the PFJ and the hip, an assessment of the foot position is necessary.

### 2.2. Foot Position

A physical examination in a standing position shows that patients with increased femoral antetorsion, in most cases, have inward pointed knees and the knee joint faces medially when the foot is in a normal position (Figure 1).

Discomfort is often noted during walking and squatting due to in-toeing. In-toeing of the foot is reported as a diagnostic clinical sign with high specificity for increased FT [4]. However, it must be noted that many of the patients with increased FT walk with a normal foot position [4]. This can lead to an underestimation or misdiagnosis of abnormal FT. Increased femoral antetorsion may be compensated by increased external tibial torsion (TT), which rotates the leg outward. These patients show medially faced knee joints with a normal foot position. They are likely to have a normal foot progression angle (FPA) due to these compensatory effects of FT and TT [4,5]. The FPA represents the angle of out-toeing of the foot compared with the line of gait progression with normal values between 5–15°. According to this, the interplay between FT and TT for the final position of the foot must be considered.

Special attention must be given to patients suffering from additional hip pain caused by posterior extra-articular ischiofemoral impingement. They try to avoid the impingement pain by active external femoral rotation. In this compensating way of walking, they experience less hip discomfort. However, on the other hand, this external rotation could provoke patellofemoral imbalance [6].

For a clinical interpretation, if a patient presents with in-toeing, there is a high probability that increased FT will co-exist. However, abnormal FT can basically be combined with any type of gait pattern of the foot. If in-toeing was used as the only diagnostic criteria for increased FT, a high percentage of patients with increased FT would be missed [4]. Therefore, measuring FT with CT scans or MRI is strongly recommended in all patients with suspicion for increased femoral antetorsion, even in the absence of in-toeing of the foot.

### 2.3. Functional Impairments

FT has a direct influence on the amount of abductor strength and the gait pattern. Objective functional impairments are of special interest. People with increased femoral antetorsion often present with external rotator and hip abductor weakness. They have difficulties with functional activities such as stair climbing, standing from sitting, squatting, and jumping. In addition, the pathologic internal rotation causes lateral forces acting on the patella and with this an “increased dynamic Q angle”. This excessive lateral force vector may cause patellofemoral pain due to the increased stress (cartilage/bone) and/or lateral patellar subluxation with instability symptoms.

## 3. Abnormal Skeletal Geometry

When are bony abnormalities clinically significant?

The major argument to perform a rotational osteotomy in patients presenting with patellofemoral complaints is a significant pathologic femoral antetorsion [1,2,7]. Therefore, patients with a high suspicion of torsional abnormalities during a clinical evaluation need precise measurements of the total FT. According to the possible great clinical importance of abnormal skeletal geometry, it is necessary to have a closer look at the physiological variations and the different values of specific measurement methods.

### Measurements

The correct quantification of FT is essential for diagnosing femoral torsional abnormalities, especially for surgical decision-making and planning of corrective osteotomies [7]. CT-based, segmental multi-level CT assessment, MR-based, and three-dimensional reconstruction measurements are available to assess FT more precisely [8,9,10].

Therefore, the most important question is: What is a normal FT?

The normal range of physiologic total FT is reported between 10 to 25° [3]. Values of more than 25–30° of total femoral antetorsion are considered as increased, decreased FT is defined as <10° [1,11,12,13,14,15]. Differences between the neck (proximal), mid (diaphysal), and distal femoral torsion were established [8,9,10]. In normal controls, a negative correlation between the neck torsion and shaft torsion was found, suggesting that the internal torsion of the neck is accompanied by an external torsion of the shaft [9]. In subjects with a high-torsion, a significant increase in the neck internal torsion in combination with a high lack in the shaft external torsion was described [9]. An evaluation of the segmental torsion of the femur allows a more detailed analysis of femoral alignment [8]. All three levels contribute to the total FT [9].

The values of FT differ considerably depending on the variety of measurement methods and the location of measurement. Using CT-based measurements, the Murphy method with the most distal definition for the proximal femoral neck axis showed high values of mean FT (28°), the Lee method with the most proximal definition of the neck axis showed low values (11°). It is reported that the Murphy method most closely reflects the true anatomic FT [3]. It is crucial to state the applied method for a correct assessment of femoral torsion. Disregard of the differences among the methods could lead to a misdiagnosis in surgical planning and/or postoperative control. The same measurement methods must be used to compare pre- and postoperative conditions. The personal clinical experience is helpful to choose the preferred measurement method.

The values of tibial torsion (TT) must also be considered for a complete assessment of the lower limb torsion. This is of special interest for the clinical examination (foot position with FPA). TT is assessed by measuring the rotational angle of the proximal tibia (line along the posterior tibial plateau) relative to the distal tibia (straight line through the centers of the medial and lateral malleolus). A normal TT is defined between 25 and 40° [16]. Values of more than 40° of TT are considered as increased, decreased TT is defined as < 25°. The femorotibial index is then calculated with TT minus FT [17,18].

For a clinical interpretation, there is no clear recommendation when a correction of pathologic FT is indicated considering the wide range of physiological femoral rotation values.

According to our experience, if total femoral antetorsion exceeds the normal by 25° and no additional pathologic morphology in the PFJ is documented, femoral external rotational osteotomy should be considered, if it exceeds 30° it is recommended [1,8,11] (Table 1). This applies to all patients with long lasting symptoms around the knee joint, clear pathologic findings during a clinical evaluation, and an unsuccessful conservative treatment.

Considering the direct interplay between FT and TT, combined femoral and tibial rotational corrections at the same time should be avoided. It is recommended to correct at first one site and then, to repeat the measurements with the same method of the limb alignment. With this, over- or undercorrections could be avoided.

## 4. Rotational Osteotomy

What should we consider performing a rotational osteotomy?

### 4.1. Site of Correction

Actually, there is still a debate about the ideal site for the osteotomy, which can be performed proximally (intertrochanteric level) or distally (supracondylar level) [8]. When only the total FT was measured, no differentiation can be made for a torsional deformity located at the proximal, diaphyseal, or distal aspect of the femur. Knowing the segment of the femur in which the torsional deformity is localized could be helpful. If an osteotomy is performed at the site of the deformity, a normal anatomy could be probably better restored [8]. However, the clinical relevance of correction osteotomies performed at the site of the pathologic segment is not proven to date. Earlier, corrections were performed at the intertrochanteric level to avoid an acute angular change of the quadriceps muscle. In addition, patients with a significant increase in neck internal torsion (documented by segmental measurements) could be corrected at the proximal level considering the differences of torsion in the three segments [9]. In addition, surgeons were used to performing proximal osteotomies over a long period.

In the last two decades, supracondylar osteotomies were increasingly used to correct additional pathologies at the patellofemoral joint (trochleoplasties, distalisation of the patella, cartilage repair) or the soft tissues (MPFL reconstruction, ligament balancing) at the same time [1,7]. Meanwhile, isolated external rotational osteotomies are routinely performed at the supracondylar level without complications such as muscular problems or complaints due to the initial incongruity of the rotated bones. The major advantages of the supracondylar correction include the possibility to perform the necessary additional interventions at the same site, for example, arthroscopy, intraarticular treatments, MPFL reconstruction, and finally soft tissue balancing [1,5,7].

For clinical relevance, both intertrochanteric and supracondylar procedures externally rotate the trochlea underneath the patella, restore correct stability, decrease the compression in the lateral trochlea, and reduce patellofemoral (and hip) pain [1,5,12,17]. The site of the osteotomy is decided based on personal preference.

The rotational correction for maltorsion of the tibia is performed below the tibial tubercle in the proximal, medial, or distal tibia. The use of internal or external fixation devices is possible.

### 4.2. Surgical Technique

A lateral subvastus approach is used for the supracondylar osteotomies [1,7,12]. Two Kirschner wires are placed anteriorly 3–5 cm above the metaphysis to monitor the planned angle of correction. The osteotomy is performed horizontally with an oscillating saw and then the distal femur is externally rotated to correct the increased internal rotation until the two Kirschner wires are positioned as the preoperative planning has indicated. Fixation of the osteotomy is performed using a locking screw osteosynthesis plate (Figure 2a,b).

The final step consists of balancing the soft tissues around the PFJ. Postoperatively, partial weight bearing with 20 kg for 6 weeks is recommended.

### 4.3. Combined Bony Deformities

Patients with patellofemoral complaints may have complex bony deformities in combination with torsional abnormalities, varus-/valgus deviation, and dysplastic trochlea [7,12,15]. In these cases, all underlying pathologies should be addressed to restore the correct limb alignment [7,12]. The same lateral approach is used, in which the individually adapted surgical procedure is performed.

### 4.4. Clinical Experience

Femoral rotational osteotomies to correct deformities have been increasingly performed with encouraging results [3,19,20]. Distal femoral derotational osteotomies are an excellent treatment option in patients with patellofemoral instability or pain [1,5,7,12]. Moreover, combined surgical interventions showed good subjective and objective functional results in most cases [5,7,12]. A significant decrease in pain, increased comfort when walking and squatting, as well as an improved function were noted [7].

The strongest argument to perform an external rotation osteotomy is the fact that it corrects the underlying pathology and restores the horizontal limb alignment (Table 2).

It improves not only patellofemoral kinematics and with this patellar stability, but also the involved muscle forces, soft tissue restraints, and hip movements. Good subjective and objective functional results are reported in most cases.

## 5. Reasons for Surgical Failures

What may lead to an unsuccessful clinical outcome?

We have seen that many surgeons hesitate or refuse to perform rotational osteotomies, although a clinical assessment and imaging documented the pathologic femoral antetorsion. The major concerns are summarized in Table 3. In these cases, other surgical interventions were preferred.

Considering this, we observed numerous patients with remaining complaints after variable surgical interventions to treat patellofemoral problems. The precise assessment of the skeletal morphology depicted in some of these cases of pathologic femoral antetorsion values as the single or most important cause for the patellofemoral problems, which was not addressed. This means that these patients had a clear indication for derotational osteotomy, but other interventions were performed. Only the secondary external rotation osteotomy on the supracondylar level could finally resolve the problem. Occasionally, it was even necessary to reverse the primary procedures at revision.

Rather than the correction of a pathologic rotation, other interventions, such as MPFL reconstruction, medial soft tissue plication, medialization osteotomy of the tibial tubercle, or raising of the lateral trochlea, were performed. Such attempts to compensate the primary problem without a precise clinical evaluation and the corresponding imaging measurements may not only result in unsuccessful outcomes, but may also cause additional complaints. Medialization of the tibial tubercle leads to increased external tibial torsion and may therefore exacerbate the symptoms [5]. MPFL reconstructions risk pulling the patella too much to the medial with increased compression. Raising of the lateral condyle may cause lateral hypercompression.

It is most important that the chosen surgical procedure aims to eliminate the cause of the patellofemoral problem. Therefore, a detailed analysis to identify the relevant factors is crucial. The surgical intervention should aim to regain mostly normal anatomic and biomechanical conditions. Lately, abnormal femoral torsion was depicted in about 23% of patients suffering from patellar instability, thus representing an important risk factor with clinical implications [9].

We observed also patients with multiple causes and factors for patellofemoral complaints.

Increased femoral antetorsion can be combined with trochlear dysplasia, insufficiency of the MPFL, patella alta, pathologic tibial torsion, increased tibial tuberosity-trochlear groove distance, pathologic coronal alignment (valgus/varus), and/or different hip abnormalities [5,7,12,15]. Occasionally, only one risk factor for patellar instability is addressed during surgery (Figure 3a–c). This may lead to persisting complaints.

For clinical relevance, derotational osteotomy should be considered in the first place in patients with increased femoral antetorsion, other compensatory interventions are avoided whenever possible. If multiple risk factors for patellar instability or pain are documented, then a meticulous analysis of what should be corrected is necessary. We have learned that sometimes combined interventions are necessary.

## 6. Alternatives

Are there alternatives for rotational osteotomies?

The association of the described functional impairments offers the possibility of structured training to improve the functional performance. A dynamic hip strength and power might improve the objective function of the whole limb and the patellofemoral tracking. Therefore, rehabilitation should focus on hip extension, hip external rotation, and hip abduction to control femoral rotation movements. This includes progressive resistance training for strength, power, and muscular endurance. In addition, a core muscle control with trunk stability must be addressed.

## 7. Conclusions and Observations

Rotational osteotomies are indicated for those patients with a significant pathologic total femoral torsion and persistent patellofemoral complaints after failed conservative therapy over a long period. A supracondylar external rotation osteotomy is recommended if the total femoral antetorsion value exceeds 25°.

Distal femoral external rotation osteotomy is an individually adapted surgical procedure for restoring the horizontal limb alignment. It improves patellar stability and patellofemoral kinematics and yields good subjective and objective functional results.

## Figures and Tables

**Figure 1 jcm-10-00474-f001:**
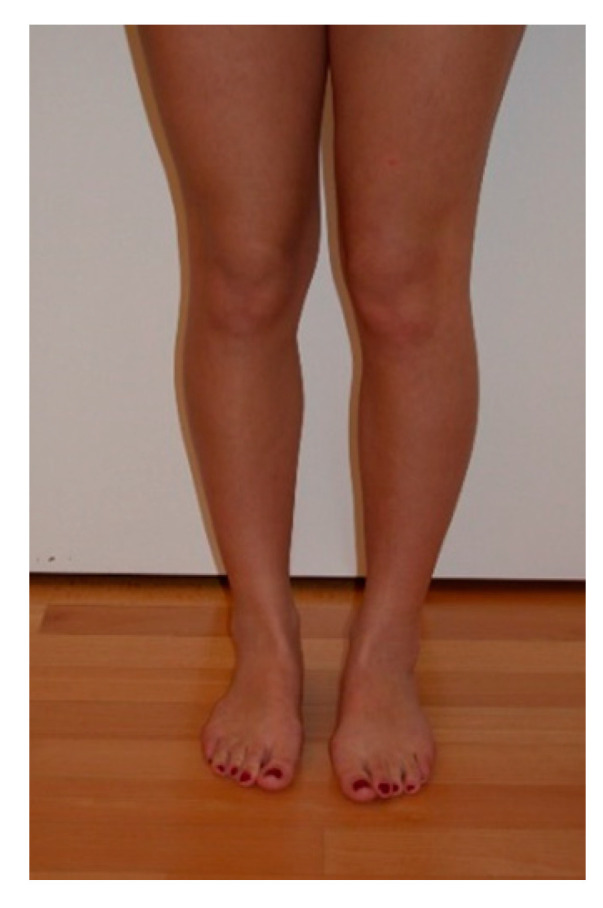
View from the front visualizing the squinting patellae with increased femoral antetorsion, the left side more than the right side.

**Figure 2 jcm-10-00474-f002:**
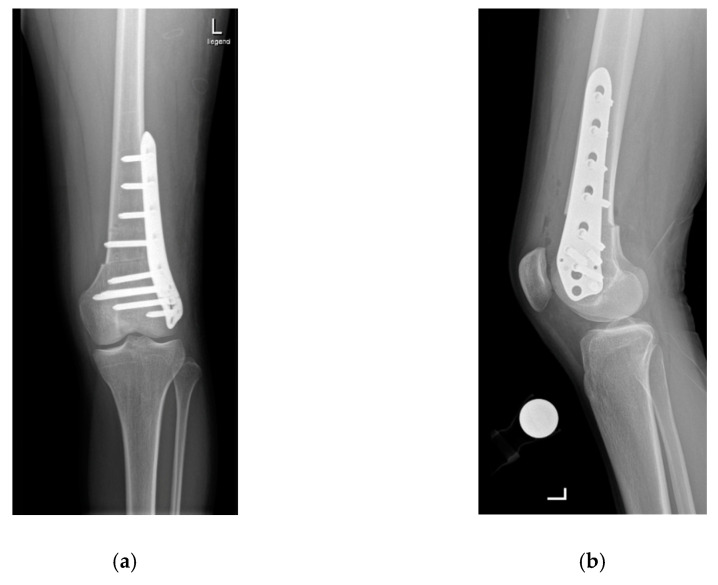
Anteroposterior (**a**) and sagittal (**b**) radiographs showing the situation after supracondylar rotational osteotomy.

**Figure 3 jcm-10-00474-f003:**
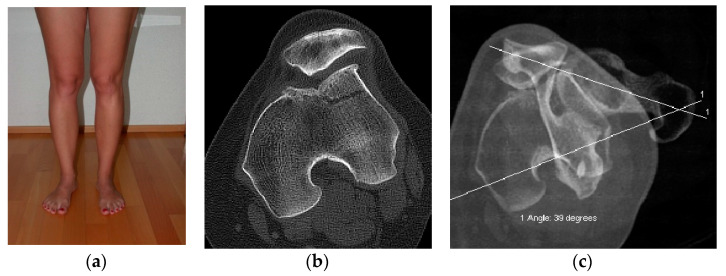
Postoperative standing examination (**a**) View from the front visualizing increased femoral antetorsion on the left side after deepening trochleoplasty and persisting lateral patellar subluxation; (**b**) axial computed tomography showing the situation after deepening trochleoplasty with persisting lateral patella subluxation (the same patient as in Figure 3a); (**c**) Rotational study with computed tomography showing increased femoral antetorsion of 39° (the same patient as in Figure 3a,b).

**Table 1 jcm-10-00474-t001:** Indications for femoral external rotation osteotomy.

Clear correlation between increased femoral antetorsion and patellofemoral complaints.
Increased total femoral antetorsion documented by imaging.
Persisting symptoms with failed conservative treatment over a long period.

**Table 2 jcm-10-00474-t002:** Main arguments for rotational osteotomies.

Rotational osteotomies correct pathologic femoral torsion.
Clinical experience with encouraging results after femoral derotation osteotomies.
Good clinical and functional results after secondary rotational osteotomy in failures after other patellofemoral interventions.

**Table 3 jcm-10-00474-t003:** Concerns about rotational osteotomies.

Different available measurement methods with a wide range of values.
Differences between neck, mid, and distal femoral torsion and their contribution to the total femoral torsion.
Unclear recommendation for proximal or distal osteotomy.
Invasive surgery with locking plate and plate removal.
Long rehabilitation time.

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
