# Peer review of "Reflections on Rotational Osteotomies around the Patellofemoral Joint"

_jcm, 2021, doi:10.3390/jcm10030474_

Round 1

Reviewer 1 Report

Dear Sirs,

An excellent over view from an experienced author.

From my point of view, to improve the paper I would like to see a section on how tibial torsion is measured, what is considered abnormal, how is is corrected and how this interplays with normal and abnormal femoral rotation. 

I think that if this section is included then it would complete what is an excellent and interesting piece of work worthy of publication.

Author Response

Thank you for your time.

The measurement method for tibial torsion is inserted directly into the text.

Normal and pathologic values of TT are already described in the text.

The site for correction of tibial malrotation is inserted directly in the text.

Reviewer 2 Report

As the article is an opinion piece , it makes it hard to judge . It does a pretty good review of the topic and discusses the arguments in a cohesive manner . It needs some minor adjustments in grammar but they are easily corrected . If your journal accepts opinion pieces , I would accept it as I think it was quite interesting and well done . 

Author Response

Thank you.

This paper was intended to describe personal observations and thoughts.

Reviewer 3 Report

This is an opinion about the rotational osteotomies to correct torsional abnormalities of the femur affecting the patella instability of patellofemoral complaints. I agree with the importance of measuring the rotational alignment in the treatment of patellofemoral disease. This is an interesting topic, and I recommend minor Revision.

Page 3, “There-fore, measuring FT with CT scans or MRI is strongly recommended in all patients with suspicion for increased femoral antetorsion, even in the absence of in-toeing of the foot.”

-     Why do we need an MRI to measure femoral anteversion? The most important thing is the physical examination. First, we can have information about the femoral antetorsion by checking the increased hip internal rotation. Second, we can check the femoral anteversion through the trochanteric palpation method (Davids, J.R. Assessment of Femoral Anteversion in Children With Cerebral Palsy: Accuracy of the Trochanteric Prominence Angle Test. J Pediatr Orthop 2002, 22, 173-178). Third, we can measure femoral anteversion and tibia torsion using a CT scan.

Page 8, “A supracondylar external rotation osteotomy is recommended if total femoral antetorsion value exceeds 25°.”

-     Why do the authors recommend 25 degrees? In my experience, patients with cerebral palsy often have increased femoral anteversion more than 25 degrees, but they do not have patellar instability. I also experienced individuals with increased femoral anteversion who did not have patellofemoral instability. Furthermore, we should check genu valgum with femoral anteversion. In this article, the evidence for femoral osteotomy recommendation in patients with more than 25 degrees femoral antetorsion is just normal range (10-25). I recommend authors should modify the conclusion. I think the author’s opinion is that “not only soft tissue structure and genu valgum (coronal alignment) but also femoral antetorsion (axial alignment) should be considered as a risk factor for patella instability.”. 

Author Response

Thank you for your time to review this paper.

I offer the following answers to your comments:

  • Page 3, “There-fore, measuring FT with CT scans or MRI is strongly recommended in all patients with suspicion for increased femoral antetorsion, even in the absence of in-toeing of the foot.”

1st: I totally agree with you that the clinical examination, as you mentioned, is most important. I focused on this in the text under 2.1.General examination. MR or CT measurement is needed to document the exact amount of increased femoral rotation, especially when considering rotational correction.This is mandatory for preoperative planning. 

2nd: Some authors prefer MR measurement to avoid radiation exposure from computed tomography. Especially in younger patients this makes sense.

  • Page 8, recommended 25°

Most papers in the literature (see references) use this cut-off value as indication to perform surgical correction.This applies to the majority of people. But of Cours I agree that patients with cerebral palsy often have increased femoral ante torsion of more than 25 degrees.

  • Furthermore, we should check genu valgum with femoral anteversion. 

I agree with your comment. This paper focused in first line on rotational deformities and the coronal plane was not specifically involved. Of course genu valgum may have an important impact on patellar instability and femoral rotation. I these cases both must be addressed for surgical correction. The main focus of this paper was on rotational abnormalities.

  • I recommend authors should modify the conclusion. 

With this I don't agree. I think (and this corresponds with the literature) that increased femoral antetorsion is a more significant factor for patellar instability than genu valgum. But of course, genu valgum is an important secondary risk factor for instability, especially in combination with increased femoral antetorsion.